# Estimating the Cultural Value of Wild Animals in the Qinling Mountains, China: A Choice Experiment

**DOI:** 10.3390/ani10122422

**Published:** 2020-12-17

**Authors:** Yilei Hou, Tianle Liu, Zheng Zhao, Yali Wen

**Affiliations:** 1School of Economics and Management, Beijing Forestry University, Beijing 100083, China; houyilei@bjfu.edu.cn (Y.H.); m13051856877@163.com (T.L.); 2College of Tourism, Shanghai Normal University, Shanghai 200234, China; zzshnu@shnu.edu.cn

**Keywords:** wild animals, cultural value, choice experiment, biodiversity conservation, Qinling Mountains

## Abstract

**Simple Summary:**

The presence of wildlife makes an important contribution to ecosystem services, and in the hot spots of biodiversity conservation, the results of wild animal value evaluation can provide a basis for the formulation of ecological compensation standards. Applying the choice experiment method, we assessed the value of wild animals in the surrounding areas of the Changqing National Nature Reserve in China from the tourist perspective. The results of our field survey suggest that the cultural value of wild animals, measured in terms of willingness-to-pay, is significantly higher than that of the infrastructure and other natural landscape features. These findings can provide a basis for the rational allocation of resources for wild animal conservation.

**Abstract:**

In this study, we use the choice experiment method to evaluate the value of wild animals in the surrounding areas of the Changqing National Nature Reserve in China. For areas focused on biodiversity conservation, the results of wild animal value evaluation can provide a basis for the formulation of local ecological compensation standards. We identified wild animals, natural landscape features, infrastructure and facilities, and ticket price as major attributes determining the utility and value of the reserve area. The results of our field survey suggest that the cultural value of wild animals is significantly higher than the value of the infrastructure and other natural landscapes. Tourists indicated a willingness-to-pay of 31.4 Yuan to see wild animals in captivity, compared to 71.9 Yuan to see wild animals in the field. Moreover, tourists with different characteristics have different preferences for the cultural value of wildlife. Female tourists have a higher willingness-to-pay than male tourists. In addition, as age, education, and income level increase, tourists’ willingness-to-pay also increases. These conclusions can provide a reference for local government to formulate wildlife protection compensation and ecotourism ticket pricing.

## 1. Introduction

Ecosystem services are an effective way to strengthen the relationship between ecosystem functions and human well-being [1]. The Millennium Ecosystem Assessment divides ecosystem service functions into four categories: supply, regulation, cultural, and support functions [2]. Cultural ecosystem services are defined as ‘the non-material benefits people obtain from ecosystems through spiritual enrichment, cognitive development, reflection, recreation and aesthetic experience, including, e.g., knowledge systems, social relations and aesthetic values’ [3]. Compared with other ecological services, cultural services are more easily perceived and experienced by human beings, and they play an important role in increasing human welfare. A greater or lesser cultural significance can be assigned based on diverse factors, both those linked to the particular cultural features of the human group assigning it and the intrinsic features of the species [4].

Ecotourism is a responsible travel to natural areas that conserves the environment, sustains the well-being of the local people, and involves interpretation and education. It can improve the well-being of local community residents and tourists [5]; however, there is still uncertainty about whether ecotourism can improve the level of ecological protection [6]. Some studies have shown that tourists’ perceived value of the ecological environment has a certain impact on their protection behavior [7]. Therefore, it is of certain significance to study the value evaluation of tourist destinations in the process of ecotourism.

With respect to the evaluation of ecosystem services and policies pertaining to the ecological environment, a long-standing difficulty is that some products or services provided by ecosystems do not have a market that can reflect their scarcity [8]; hence, they cannot be priced. Owing to a lack of trading markets and the difficulty of defining the property rights to these products or services (e.g., air quality, biodiversity, and climate-regulating ecosystem services), natural resources have always been over-exploited [9,10,11,12]. The absence of the trading object produces the same result. For example, the goal of intergenerational resource allocation efficiency is to secure the right of future generations to use the same quality and quantity of natural capital as current generations [13]. However, future generations cannot negotiate with current generations to determine the appropriate discount rate as they cannot travel from the future to the present. Based on this, the choice experiment provides new ideas and methods for the pricing of ecological products and services [14,15]. Economic evaluation of ecosystem services has many benefits but also has considerable downsides. Due to the definition of ecosystem service or the evaluation method, the evaluation results of ecosystem economic value are often controversial.

Since the 1990s, choice experiments have been widely used in the fields of ecological environment, natural resource utilisation, and cultural value [16,17,18]. Hanley et al. studied the factors influencing the different needs of climbing sites in Scotland using four attributes [19]. Miller et al. designed a choice experiment to evaluate the cultural value of freshwater to indigenous peoples in New Zealand [20]. They identified five important local attributes of freshwater resources, including indigenous cultural value. The results show that the estimated willingness-to-pay was $40/year for individuals belonging to the Māori ethnic group, compared to $28/year for the general public [21]. Oleson et al. applied a discrete choice experiment to determine indigenous fishers’ preferences and willingness-to-pay for bequest gains from management actions in a locally managed marine area in Madagascar. They found that bequests are highly valued and important, and respondents were willing to pay a substantial portion of their income to protect ecosystems for future generations [22].

Wild animals are an active and attractive part of the ecosystem, as well as an important material resource for human development [23]. The intrinsic value of wild animals has always attracted people’s attention. They are important resources for natural development and biodiversity protection. The cultural value of wild animals plays an important role in ecotourism development, and it has become an important motive for people to participate in ecotourism activities [24,25]. There have been several studies on the direct and indirect value evaluation of wildlife [26,27], but there are few studies on the cultural value of wildlife. Utility theory posits that the utility for consumers can be regarded as the aggregate utility derived from multiple attributes [28]. In ecotourism activities, the value of wild animals is an attribute of the utility that people obtain from participating in ecotourism activities.

In this study, the cultural value of wild animals is regarded as one of the important attributes of the ecosystem service function of ecotourism destinations. Choice experiments were carried out to obtain tourists’ consumption preferences using a questionnaire survey, from which the specific cultural value of wild animals could be calculated using a logit regression model. The results suggest that the cultural value of wild animals is significantly higher than the value of the infrastructure and other natural landscapes. The research results can provide a theoretical basis and practical reference for the rational allocation of resources and compensation for wild animal conservation, for example, the ticket price of ecotourism.

## 2. Materials and Methods

### 2.1. Experimental Design and Variable Selection

The choice experiment, originally developed by Louviere and Hensher and Louviere and Woodworth [29], belongs to the statement preference method. However, it is different from the direct bidding method commonly used by the statement preference method. This method simulates a trading market with different levels of attribute combinations, so the evaluation results are relatively objective. Owing to their flexibility, choice experiments are perhaps the most appropriate method for eliciting values in complex situations involving trade-offs between multiple ecosystem services, particularly those linked to socio-cultural values [30,31,32]. As wildlife value is an ecosystem service values, it is feasible to use this method for evaluation purposes.

For a choice experiment, a questionnaire is designed to obtain information on people’s willingness-to-pay for a certain attribute [33,34,35]. Different levels of item attributes are combined, and these combinations are configured into a choice set according to the unbiased and efficient principles of statistical estimation. After comparing the options, respondents make a selection from one of the choices [36]. Therefore, the choice experiment is different from the direct bidding method used in the open or closed conditional value method (CVM) [37]. It simulates a trading market with different attribute combinations. This avoids circumstances in which respondents always answer ‘yes’, as common in CVM, and also avoids the collinearity problem present in the revealed preference method. Choice experiments can evaluate one aspect of the ecological environment and resource utilisation, while CVM is more suitable for evaluating the overall value of the ecological environment and its resources.

#### 2.1.1. Attributes and Level Identification

This study attempts to determine the value of wild animals through the evaluation of ecosystem services. When people participate in ecotourism around a conservation area, the main purpose is to enjoy the services of the ecosystem [38], and the value of wild animals is a part of the overall ecosystem service value. The choice experiment method allows us to calculate the value of this specific component.

Considering the challenges of past studies on the evaluation of ecosystem value using choice experiments and in line with the guidelines laid out in Bennet and Birol [39], several steps were involved in the design of our experiment. The attribute selection process involved a comprehensive literature review of past relevant choice experiments, as well as studies on what ecosystem services exist in China, especially in the Qinling area. Focus group discussions (n = 7) and key informant interviews (n = 20) were used to identify: (i) the ecosystem services people use when they travel to the conservation area and whether viewing wild animals are one of them; and (ii) an appropriate payment vehicle. These findings can reflect different ecosystem values which are mainly used to guide the selection of the most important attributes and explore how to describe the attributes in the survey. Based on this, we designed the attributes and levels for the choice experiment and conducted a pre-survey with area tourists. The main purpose of the pre-survey was to understand tourists’ perception of the value of the ecosystem services and determine whether they can understand the attributes and level expressions in the experiments. According to the results of the pre-survey, we adjusted the specific level and expression of each attribute. Finally, we identified wild animals, natural landscape, infrastructure and facilities, and ticket price as major attributes that determine the utility and value of tourism in and around the Changqing National Nature Reserve (Table 1).

#### 2.1.2. Experiment Design

Experimental design is the process of combining different levels of attributes into alternatives, and then combining the various alternatives into a choice set. The purpose of experimental design is to obtain as much useful information as possible using as few selection sets as possible. Experimental design includes full factorial design and partial factorial design, but full factorial design is only suitable for selective experiments with fewer attributes and levels. Partial factorial design can reduce the number of selection sets [40]. We use D-efficiency design, the most effective method of partial factorial design [41], to select the alternatives for the selection set.

According to the attributes and levels determined in this study, if a full factorial design is adopted, a total of 72 (2 × 3 × 3 × 4) combinations of the different attribute states can be produced. This study used SAS statistical software to screen 24 selection sets through D-efficiency design, and the selection sets were randomly paired to form 12 selection sets. Each of these selection sets contains one current plan and two improvement plans (Table 2). The current plan involves maintaining see wild animals in captivity, infrastructure, and the other attributes at their current levels, among which, the tourist did not participate in the payment except ticket. The improvement plans indicate that the various attributes of the tourism destination evaluation and utilisation in the survey area must reach the level of quality required by the state, and tourist participation is essential to this process.

Considering that too many choice sets may lead to psychological fatigue for the interviewed tourists, the questionnaire is divided into two versions (Questionnaire A and Questionnaire B). Each version includes six choice sets, and tourists need to choose the most preferred option out of the three options in each choice set.

To allow tourists to make more accurate answers to the experimental content, other qualitative questions were also incorporated into the questionnaire. The contents of the questionnaire included the following: personal characteristics of tourists, travel distance, travel expenses, knowledge of ecosystem service functions, understanding of the types and names of local wild animals, and satisfaction with the natural landscape and basic facilities of Changqing tourist destinations.

### 2.2. Data and Method

#### 2.2.1. Study Area

For the geographic area of interest, we selected China’s Qinling Mountains (Figure 1). Qinling is the geographical boundary between the north and south of China, as well as a clear boundary for the country’s flora and fauna [42]. In this area, the natural resources are famous for their primitive qualities, especially the uniqueness of the landscape. It is also one of the areas with the richest species diversity and ecosystem in China [5]. At the same time, Qinling, the watershed of the Yellow River and the Yangtze River, has nurtured a long history of Chinese civilisation and holds a high ecological and cultural value.

The Changqing National Nature Reserve is located on the south slope of the Qinling Mountains. It was mainly established for the protection of the giant panda, crested ibis, golden monkey, takin, and other wild animals. It is in the transitional zone between the north subtropical zone and the warm temperate zone, with an area of 30,000 ha and more than 90% forest coverage. It is also a densely distributed area of giant pandas, which has attracted extensive attention from domestic and foreign scientists. In 2004, the Changqing National Nature Reserve was selected among one of the first batches of the world’s best managed reserves (IUCN Green List). Currently, discussions on wildlife resource protection, effective utilisation, and sustainable development are becoming increasingly intense. As an important part of ecotourism, wildlife has gradually become an important factor in attracting tourists.

Huayang Town, an area of historical significance, near the Changqing National Nature Reserve is an ecotourism destination that integrates nature sightseeing, wildlife viewing, exploration, and scientific research. In recent years, it has attracted a large number of tourists, many of whom travel to the area for the purpose of viewing wildlife. Visitors can often see crested ibis, golden monkeys, and, on occasion, temporarily captive giant pandas that have been rescued by the reserve.

#### 2.2.2. Data Collection

Survey data were collected in July–September 2018. To optimise the investigation plan for the choice experiment, a preliminary survey was conducted in August 2017, and the feedback was used to understand whether the attributes and levels of the design of the experimental program were consistent with reality. The content of our questionnaire mainly includes the interviewee’s personal characteristics, tourism purpose, cognition and attitude towards wildlife conservation, and the selection set of choice experiments (see Appendix A for details). Our survey team comprised graduate students and staff from the Nature Reserves Authority who were trained socioeconomic researchers.

The survey objectives and questions were explained to the participants to minimise potential miscommunication when administering the questionnaire. Once the participants had finished the survey, they received thank-you gifts, such as souvenirs. According to the preliminary survey results, more than 95% of tourists were able to complete the wild animal cultural evaluation with the help of investigators. The research team adjusted and optimised the survey plan based on the results and officially launched the main survey in 2018. Our data collection team comprised 12 graduate students from Beijing Forestry University who are trained social researchers with rich experience in tourist surveys. These researchers have assisted us with other survey-based projects over the past decade.

To ensure that the respondents have a perceptual understanding of each attribute, we produced a picture book about the artificial breeding of wild animals, wild animals in the wild, the deterioration of natural landscapes, the improvement of natural landscapes, and better scenic facilities. The interviewer displayed the pictures so that the respondent could understand the meaning of each attribute and level. Given the time required for the interviewees to complete the experiment (each questionnaire takes about 20–30 min), we gifted a golden monkey plush toy worth 20 yuan to each interviewee to incentivise them to complete the questionnaire. Ultimately, 128 choice experimental questionnaires were received, of which 114 were valid.

#### 2.2.3. Choice Experiment Method

Theoretically, the choice experiment originated from Lancaster’s characteristic demand theory [43]. Lancaster asserted that utility does not come from the direct object of goods, but from the attributes or characteristics of goods. However, in terms of method, the choice experiment is based on the theory of random utility [44,45,46]. According to the theory of random utility, the indirect utility function for each respondent *i* (*V_ij_*) can be decomposed into two parts: one deterministic, observable component (*V_i_*) and one random and unobservable component (*ε_ij_*). It can be formulated as:(1)$$V_{ij}=V_{i}\left(x_{j},T_{j}\right)+\varepsilon_{ij},$$
where *V_ij_* is the total utility that individual *i* derives from alternative *j*, and *V_i_* is the observable utility of individual *i* derived from alternative *j*. Accordingly, ε*_ij_* is the unobservable utility that individual *i* derives from alternative *j*. Variable *x_j_* is a particular attribute of ecotourism service *x* in alternative *j*, and *T_j_* is the willingness of individual *i* to participate in alternative *j*.

The probability of individual *i* choosing alternative *j* rather than *n* in a given choice set is the probability that the random utility of alternative *j* is greater than the random utility of alternative *n*. The probability of choosing alternative *j* is [47,48]:(2)$$P_{i}=Prob\left(U_{j}>U_{n}\right)=Prob\left(V_{ij}+\varepsilon_{ij}>V_{in}+\varepsilon_{in}\right)=Prob\left(V_{ij}-V_{in}>\varepsilon_{ij}-\varepsilon_{in}\right),\;\forall j\neq m.$$

The estimation choice probabilities are determined by the distribution of the disturbance term ε. The choice experiment usually assumes that ε follows the logistic distribution, and the probability of individual *i* choosing *j* is:(3)$$P_{ij}=\frac{e^{V_{ij}}}{\sum_{j}e^{V_{in}}}.$$

In a choice experiment, respondents are presented with a series of alternatives, differing in terms of attributes and levels, and asked to choose their most preferred. Choosing the right attributes and levels is the key to assessing whether the experiment can accurately price because the attributes determine the utility and value of the goods. The purpose of presenting different levels for the selected attributes is to calculate the marginal substitution rate between attributes.

If we assume that the utility obtained by consumers from the attributes of goods is additive, *V* in (2) can be expressed as:(4)$$V=\overset{n}{\underset{k=1}{\sum }}\beta_{k}x_{k},$$
where *x_k_* is the *k*th attribute of the article, *β_k_* is the marginal utility of the attribute, and the marginal substitution rate of attribute *X*_1_ and attribute *X*_2_ is:(5)$$MRS_{12}=\frac{\partial V/\partial x_{1}}{\partial V/\partial x_{2}}=\frac{\beta_{1}}{\beta_{2}}.$$

If *β_cost_* is the marginal utility of cost, then the marginal willingness-to-pay of attributes *x_k_* is:(6)$$WTP=-\frac{\beta_{k}}{\beta_{cost}}.$$

A choice experiment is usually characterised by six key stages: selection of attributes, assignment of levels, choice of experimental design, construction of choice sets, measurement of preferences, and estimation procedure. In the experiment, individual preferences can be uncovered through surveys that ask respondents to select their preferred choice.

In this study, the Changqing National Nature Reserve is the research object, and the tourists in this area are the survey objects. Of the 128 questionnaires sent out, 114 valid questionnaires were recovered, representing an effective rate of 89.06%. Definitions and statistical measurements of the key variables are summarised in Table 3.

In this study, a multinomial logit (MNL) model is used to analyse the data and estimate the marginal substitution rate between attributes and the cultural value of wildlife in the reserve. Model 1 uses a linear regression model (OLS) and assumes that respondents’ choice of alternatives is only related to the level of attributes affecting the cultural value of wildlife. Model 2 uses an MNL model and assumes that the decision-making is also affected by personal characteristics, such as gender, age, education, personal monthly income, religious beliefs, and permanent residence, assigning values to each (as shown in Table 3). Given that the same respondent produces 18 observations, to avoid repeating the same observation 18 times, the cross-terms are also included in the model calculation. The utility functions of Models 1 and 2 are given by Equations (7) and (8), respectively. See Table 3 for the variables and their explanations. Where the variable reference item has taken a zero value,
Y = β1 ticket + β2 wild animal + β3 scenery + β4 infra + β5 const + ε,(7)
Y = β1 ticket + β2 animal + β3 scenery + β4 infra + β5 const + β6 animal*gen + β7animal*age + β8 animal*edu + β9animal*inc + β10 animal*rel + β10 animal*res + ε.(8)

## 3. Results and Discussion

### 3.1. Descriptive Statistical Analysis of Tourist Demographics

This study used Excel and SPSS20.0 to perform descriptive statistical analysis on tourists’ social, economic, and personal characteristics data. The results of statistical analysis is shown in Table 4.

According to the survey results, 78.07% of the tourists, especially those with children, travel to the area for the purpose of seeing wild animals. Most of them said in the interview that through this tour, they hope their children can become more familiar with some of the wild animals and have an educational experience. Therefore, wildlife is the most important factor to attract tourists to Huayang Town. The results further illustrate that wildlife has played a certain role in promoting ecotourism, and the protection of wildlife has brought certain social benefits [6].

According to the survey data presented in Figure 2, the vast majority of visitors interviewed from Changqing National Nature Reserve come from Shaanxi Province, accounting for 91%. Conversations with tourists have highlighted that tourism advertising has a prominent place in local television programming with the province, while the television station of the Central Committee of the Communist Party of China does not offer the same visibility to potential tourists from the rest of the country. Therefore, the visiting tourists mainly originate from Shaanxi Province, and there is no obvious regularity in the distribution of other tourists. Among the 104 tourists from Shaanxi Province, 64 tourists from Hanzhong City, where Huayang Village is located, account for more than half of the total number of tourists interviewed (54.1%). In terms of number of tourists, Hanzhong is followed by the cities of Xi’an and Xianyang, accounting for approximately 27% and 6%, respectively, of the tourists interviewed.

### 3.2. Impact of Different Atrributes on Value Estimation

The estimation results of Models 1 and 2 are shown in Table 5. It can be seen from Model 1 that, with the exception of the explanatory variable of wildlife under the status quo condition, all explanatory variables are significant at the 1% level. In addition, the variance inflation factors (VIF) of all the explanatory variables are less than 10, indicating that there is no autocorrelation or multicollinearity [49].

First, ticket price reduces the tourism value of the reserve and reduces the utility for tourists; that is, its marginal utility is negative, which is consistent with the expected result. In fact, tourists are not satisfied with ticket prices, especially with respect to the price settings of entertainment facilities. For example, they do not think there are many children’s entertainment. High ticket prices will reduce the utility of opportunities for people to appreciate wild animals, indicating a substitution effect between tickets and wild animals.

Second, compared with seeing no wild animals at all, seeing wild animals in artificial environments and seeing wild animals in the field will increase the utility of viewing wild animals for tourists. As one of the main motivations for tourists to travel to the area is to see wild animals, whether they see animals raised in captivity or in the wild will have a huge effect. The survey found that, in visiting the reserve, tourists gain a certain understanding of biodiversity conservation and local folk culture, which provides a significant educational experience with respect to the natural environment.

Third, infrastructure improvements increase the value of ecological products and the utility for tourists. Tourists indicated that they are very satisfied with the abundance and experience of outdoor projects in the scenic area, such as the planning and design of the scenic area, roads, traffic sign design, and road conditions. However, at the same time, there is some dissatisfaction with the current state of infrastructure construction. Therefore, if the infrastructure is improved, it will attract more tourists and encourage a large number of repeat visits to the area. This result shows that although people participate in ecotourism mainly to see wild animals, there is also high demand for other entertainment facilities. While this can lead to the development of other ecotourism attractions, it does not necessarily address the direct protection of those assets, as it is prone to market failure [50].

### 3.3. Impact of Tourists’ Socio-Economic Characteristics on Value Estimation

In terms of socio-economic characteristics, tourists’ gender, age, education level, average monthly income, religious beliefs, and whether they live in the local area are all important factors that affect their willingness to pay for tourism experiences (e.g., entry tickets, conservation fees, etc.). The results of Model 2 show that the addition of the six socio-economic characteristics as variables does not affect ticket prices, regardless of whether other natural environments maintain the status quo, other natural landscapes become better, or the utility direction and significance of infrastructure improvements are improved. However, from these results, we can see that tourists with different characteristics have different preferences for the cultural value of wild animals.

Gender has a significant impact on tourists’ willingness-to-pay. Female tourists are more willing to pay than male tourists, and they also demonstrate a stronger interest in seeing wild animals. In the survey, it was found that most of the female tourists brought their children to see the animals, hoping to supplement their nature education. Therefore, they may be more willing to pay for wildlife conservation. Male tourists pay more attention to leisure and entertainment. Many male tourists say that they are just to bring their family out. They travel here more for family services, such as driving.

Age is also an important factor. Older tourists have less knowledge of nature and wild animals, and they are less interested in participating in tourism activities. Older people pay more attention to the natural scenery of the area. Young people, especially children, are extremely interested in wild animals. They are more knowledgeable about wild animals, and most of them can accurately name the local rare wild animals.

Tourists with higher levels of educational attainment are more likely to pay for wild animals in the reserve. Given the various information media, it is easier for them to understand the rarity of these animals and the importance of protecting them. Income also significantly affects willingness-to-pay. Tourists with higher incomes are more willing to pay a certain fee for wild animals.

Religious beliefs also have a significant impact on tourists’ willingness-to-pay, and the coefficient of the regression result is negative, indicating that compared with people without any particular set of religious beliefs, religious tourists have a lower willingness-to-pay. In the survey, it was found that Buddhist tourists believe that there are many immortals found in nature, even wild animals are believed to be an incarnation of immortals, so they think that human beings should take the initiative to protect wild animals. In their view, these are things that must be done and should not be protected by the way of payment, which is a kind of remedial action after the event. This is the result of the special ideological influence of Buddhism practised in Chinese. Therefore, the Buddhist tourists’ willingness-to-pay is a little bit lower than others.

Compared to people living in the surrounding areas of the reserve, non-local tourists are more willing to pay for wild animals. This is mainly because the locals are already familiar with these animals, but the non-local tourists have fewer opportunities to see them in their normal lives. Therefore, for non-locals, the utility of travelling to the reserve to see wild animals is clearly higher and, thus, the evaluation result is also higher.

### 3.4. Results of Tourists’ Evaluation of the Value of Wild Animals

The coefficient of the explanatory variable was not significant, so the fit and explanatory effect of Model 2 was better. Therefore, Model 2 is used to calculate tourists’ willingness-to-pay for wild animals and evaluate the value of wild animals in the study area.

As shown in Equation (8), the β value obtained by regression represents the marginal utility brought by the corresponding attribute level. For example, β2 = 0.022 in Model 1; when other conditions are the same, going from not seeing wild animals to seeing wild animals raised in captivity can increase the utility for tourists by 0.022 units. According to the above description, Model 2 was used to calculate the cultural value of wild animals. According to the ticket price coefficient β1, the constant coefficient of the number of wild animals β2, and the increase coefficient of the number of wild animals β3, the willingness-to-pay is
WTP_a-anima l_ = −β2/β1 = 31.4,(9)
WTP_f-anima l_ = −β3/β1 = 71.9.(10)

As can be seen from Table 5, before the tourists’ personal characteristics are added as variables (Model 1), the estimated willingness-to-pay for wild animals is low, and if the current situation is maintained or seeing animals in the wild, the tourists’ willingness-to-pay is valued at 1.1 yuan and 5.15 yuan, respectively. Tourists’ willingness-to-pay for wild animals is lower than their willingness-to-pay for infrastructure improvements and other natural landscape improvements. After adding the six personal characteristics as variables (Model 2), the willingness-to-pay for wild animals significantly increases. The results show that, given the alternative of not seeing wild animals, tourists to the Changqing National Nature Reserve are willing to pay 31.4 yuan to see wild animals in captivity and 71.9 yuan to see wild animals in the field. This reflects that tourists believe that the value of wild animals in the wild is higher than that of those raised in captivity. In fact, most tourists who travel to the area can only see some of the artificially fed wild animals. The crested ibis, the primary rare species in this area, has a breeding base, so it is possible to see artificially reared ibis, as well as wild crested ibis in the fields. A large number of golden monkeys are based in the nearby mountains, and, where the tourist area meets the base of the mountain, visitors can see the monkeys that are drawn to this area. It is harder for tourists to see giant pandas and takins other than those rescued in the wild are temporarily housed at rescue stations within the reserve. However, some travellers who climb to sparsely populated places above 3000 m above sea level can occasionally see giant pandas and takins in the wild. In the survey, many tourists said that, when they saw the wild animals in captivity, they are satisfied just knowing that the species still exists. This shows that people have a high awareness of the intrinsic value of wild animals.

Similarly, the willingness-to-pay for maintaining the current natural landscape, improving the natural landscape, and improving infrastructure are 14.2 yuan, 9.9 yuan, and 21.1 yuan, respectively. Tourists who travel around the reserve have a certain demand for enjoying nature and other leisure and entertainment activities in the scenic area in addition to watching wild animals. Therefore, in general, the Changqing National Nature Reserve has a high recreational value, and, as the main feature of the reserve, the cultural value of the wildlife resources is indeed significantly higher than that of the natural landscape and scenic infrastructure.

Although certain wild animals, such as giant pandas, may not be seen in the wild in this reserve, they are still an important factor in attracting tourists. In recent years, in the process of advancing its ecological civilisation, China has achieved good results with respect to educating the public on the natural environment. The national awareness of nature conservation has greatly improved, and people’s tourism activities around nature reserves have greatly increased. At the same time, in scenic ecotourism sites, rich in biodiversity, such as nature reserves, wild animals have a higher cultural value [51], and it is not surprising that tourists have a higher willingness-to-pay for viewing wild animals. The tourists’ willingness-to-pay calculated here is actually the consumer surplus generated by tourists’ tourism activities [52], which can actually provide a basis for wildlife protection compensation.

## 4. Conclusions

In this study, the choice experiment method was used to evaluate the cultural value of wild animals. In designing the experiment, we identified wild animals, other natural landscapes, infrastructure and facilities, and ticket price as the major attributes determining the utility and value of the Changqing National Nature Reserve. The interviewees compared the options in each selection set and provided their preferred choices. Additionally, through regression analysis, visitors’ preferences could be used to determine their willingness-to-pay, which can be regarded as the cultural value of wild animals.

First, the results of the experiments and regression analysis show that the cultural value of wildlife is indeed significantly higher than the value of the scenic infrastructure and other natural landscapes. Relative to a decrease in the number of wild animals, tourists in the Changqing National Nature Reserve are willing to pay 31.4 yuan to maintain the status of the wild animals, but they are willing to pay 71.9 yuan to protect the wild animals and preserve the ability to see them in the wild. Therefore, the area surrounding the reserve has high recreational value owing to its rich biodiversity.

Second, tourists with different personal characteristics have different preferences for the cultural value of wildlife. Gender, age, education level, income, religious beliefs, and residence are all relevant socio-economic characteristics for this analysis. Female tourists are more willing to pay than male tourists. In addition, as age, education, and income level increase, tourists’ willingness-to-pay also increases. Compared with locals, non-local tourists have a higher willingness-to-pay.

Third, tourists are willing to pay 71.9 yuan for wild animals in tourist destinations. In fact, this can be regarded as the willingness of tourists to pay for the protection of wild animals, as well as the embodiment of the consumer surplus present in the consumption behaviour of tourists. This result, combined with the actual situation, can provide a reference for the local formulation of wildlife protection compensation standards and related policies.

Although this study evaluated the cultural value of wild animals around the reserve through experimental methods, the evaluation methods and results are relatively objective. However, it is limited to the survey of scenic spots around a protected area, and the sample size has certain limitations. In the future, the scope of research might be expanded to cover the scenic areas surrounding other similar nature reserves in the Qinling Mountains.

## Figures and Tables

**Figure 1 animals-10-02422-f001:**
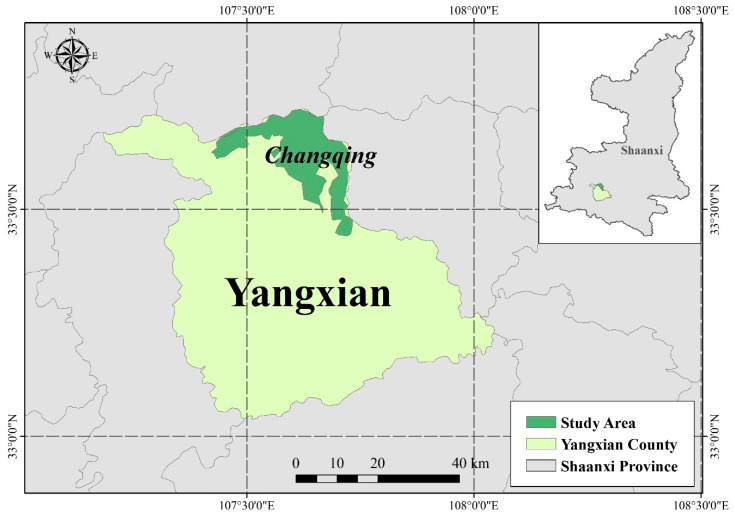
Study area.

**Figure 2 animals-10-02422-f002:**
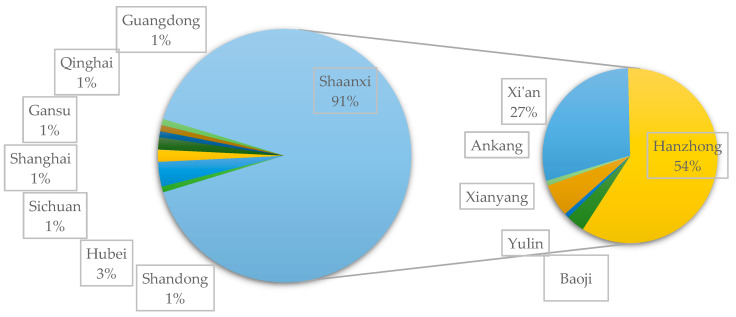
The origin of tourists to Huayang Town.

**Table 1 animals-10-02422-t001:** Description of attributes and their levels.

Attribute	Description	Levels
wild animals	Wild animals are the most important factor in attracting tourists to the area; tourists can see various wild animals raised in captivity, such as giant pandas (*Ailuropoda melanoleuca*), golden monkeys (Rhinopithecus), and crested ibis (*Nipponia nippon*), which can occasionally be seen in the wild.	1 = cannot see wild animals
		2 = see wild animals in captivity (status quo)
		3 = see animals in the wild
natural landscape	The natural scenery and other attractions in the surrounding area are also important factors that attract tourists; the better these landscapes, the higher the value of the area’s ecological services.	1 = worse natural landscape
		2 = status quo
		3 = better natural landscape
infrastructure and facilities	Food and beverage services, accommodation, and road conditions in the area also affect tourists’ decision to visit; the better these facilities, the higher the value of the scenic location.	1 = worse development of infrastructure facilities
		2 = status quo
		3 = better development of infrastructure facilities
ticket price	Tickets prices have a negative effect; the marginal substitution rate with other attributes is negative.	1 = 18.13 USD
		2 = 20.40 USD (status quo)
		3 = 22.67 USD
		4 = 24.93 USD

The amount is converted according to the average exchange rate in 2018: 1 US dollar to 6.6174 yuan (All dollar conversions in this paper are based on this exchange rate).

**Table 2 animals-10-02422-t002:** Examples of selection set in questionnaire.

Selection Set	Wild Animals	Natural Landscape	Infrastructure and Facilities	Ticket Price	I Choose
Route 1 (Status quo)	see wild animals in captivity	status quo	status quo	18.13 USD	□
Route 2 (Improvement plan)	see wild animals in captivity	status quo	better development of infrastructure facilities	22.67 USD	□
Route 3 (Improvement plan)	cannot see wild animals	better natural landscape	status quo	20.40 USD	□

**Table 3 animals-10-02422-t003:** Variable definitions and measurement.

Variables		Definition	State Level	Obs	Mean	Std. Dev
Dependent variable	Y	scheme improvement or not	1 = improvement programme;	2052	0.46	0.50
			0 = status quo scenario			
Constant term	const	programme participation	1 = participate in at least one of the improvement programmes;	2052	0.49	0.50
			0 = do not participate in any improvement programme			
Attribute argument	Z1	wild animals	1 = cannot see wild animals (n-animals);	2052	1.83	0.42
			2 = see wild animals being in captivity (a-animal);			
			3 = see wild animals in the field (f-animal)			
	Z2	natural landscape	1 = worse natural landscape;	2052	1.75	0.27
			2 = status quo;			
			3 = better natural landscape			
	Z3	infrastructure	1 = worse development of infrastructure facilities;	2052	1.77	0.36
			2 = status quo;			
			3 = better development of infrastructure facilities			
	Z4	ticket price	1 = 120 yuan;	2052	2.03	0.55
			2 = 135 yuan;			
			3 = 150 yuan;			
			4 = 165 yuan			
Characteristic independent variable	P1	Gender	1 = male;	114	1.54	0.26
			2 = female			
	P2	Age	1 = under 19 years of age;	114	3.64	1.24
			2 = 20–29 years;			
			3 = 30–39 years;			
			4 = 40–49 years;			
			5 = 50–59 years;			
			6 = 60 years and over			
	P3	education level	1 = primary school and below;	114	4.61	1.33
			2 = junior high school;			
			3 = high school (vocational);			
			4 = specialist;			
			5 = undergraduate;			
			6 = master and above			
	P4	income (personal monthly)	1 = less than 2000 yuan;	114	2.96	3.17
			2 = 2001–4000 yuan;			
			3 = 4001–6000 yuan;			
			4 = 6001–8000 yuan;			
			5 = 8001–10,000 yuan;			
			6 = 10,001–15,000 yuan			
			7 = 15,001–20,000 yuan;			
			8 = above 20,000 yuan			
	P5	Religion	1 = none;	114	1.05	0.53
			2 = Buddhism;			
			3 = Christianity;			
			4 = Islam;			
			5 = Taoism;			
			6 = other			
	P6	Residence	1 = local;	114	1.59	0.78
			2 = not local			

**Table 4 animals-10-02422-t004:** Sample structure distribution.

Basic Features	Classification	Frequency (Person)	Proportion (%)
Sex	Male	37	32.46
	Female	77	67.54
Age (years)	≤19	23	18.11
	20–29	26	20.47
	30–39	29	22.83
	40–49	26	20.47
	50–59	14	11.02
	≥60	9	7.09
Education Level	Primary school and below	1	0.81
	Junior middle school	18	14.63
	High school (vocational)	30	24.39
	Speciality	27	21.95
	Undergraduate	40	32.52
	Master and above	7	5.69
Current personal monthly income (USD)	≤301	30	24.79
	302–604	46	38.02
	605–907	26	21.49
	908–1209	7	5.79
	1210–1511	6	4.96
	1512–2267	2	1.65
	2268–3022	2	1.65
	≥3022	2	1.65
Religious belief	None	91	79.84
	Buddhism	20	17.74
	Christianity	0	0
	Islam	0	0
	Taoism	0	0
	Other	3	2.42
Residence	Local	47	41.23
	Not local	67	58.77

**Table 5 animals-10-02422-t005:** Empirical estimation results of Model 1 and Model 2.

Estimator	Model 1		Model 2	
	Coefficient	Standard Error	Coefficient	Standard Error
Ticket	−0.020 ***	0.057	−0.010 ***	0.055
a-animal	0.022	0.550	0.314 ***	0.497
f-animal	0.103 **	0.011	0.719 ***	0.009
s-scenery	0.123 ***	0.168	0.142 ***	0.117
b-scenery	0.184 ***	0.173	0.099 ***	0.154
b-infra	0.200 ***	0.013	0.211 ***	0.008
a-animal*gender			−0.092 **	0.493
f-animal*gender			0.081 ***	0.022
a-animal*age			0.011 ***	0.175
f-animal*age			0.005 ***	0.168
a-animal*education			0.013 ***	0.026
f-animal*education			0.038 ***	0.015
a-animal*income			0.033 ***	0.091
f-animal*income			0.073 ***	0.143
a-animal*religion			−0.007 ***	0.196
f-animal*religion			−0.017 ***	0.077
a-animal*residence			0.002 **	0.152
f-animal*residence			0.043 ***	0.299

(1) The regression result of the attribute variable is not added to the interactions, so the marginal coefficients can be explained directly. (2) Considering the number of parameters in one regression and to reduce collinearity, the interactions are included in the explanatory variables. (3) The sign of the estimated standard deviations is irrelevant; they should be interpreted as being positive. (4) ***, **, and * denote statistical significance at the 1%, 5%, and 10% levels, respectively.

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
