# Peer review of "Estimating the Cultural Value of Wild Animals in the Qinling Mountains, China: A Choice Experiment"

_animals, 2020, doi:10.3390/ani10122422_

Round 1
Reviewer 1 Report
Thank You for the opportunity to review this interesting paper. I have some suggestions with can improve the manuscript.
In table 1 I recommend to add the Latin names of the animals.
In table 1, 2 and in the text I recommend to change yuan to USD, because it is the most known currency in the world.
The figure 2 is very difficult to understand without detailed description in the figure caption (which should be in the same page).
Author Response
We appreciate the comments from the reviewer. We have itemized all comments and addressed them individually as follows.
Comment 1: In table 1 I recommend to add the Latin names of the animals.
Response 1: Thank you for your comments. We have added the Latin names of the animals in Table 1 as follows:
giant pandas(Ailuropoda melanoleuca), golden monkeys(Rhinopithecus), and crested ibis(Nipponia nippon).
Comment 2: In table 1, 2 and in the text I recommend to change yuan to USD, because it is the most known currency in the world.
Response 2: Thank you for your comments. We have change yuan to USD In table 1, 2 and in the text.
Table 1. Description of attributes and their levels.
|
Attribute |
Description |
Levels |
|
wild animals |
Wild animals are the most important factor in attracting tourists to the area; tourists can see various wild animals raised in captivity, such as giant pandas (Ailuropoda melanoleuca), golden monkeys (Rhinopithecus), and crested ibis (Nipponia nippon), which can occasionally be seen in the wild. |
1 = cannot see wild animals 2 = see wild animals in captivity (status quo) 3 = see animals in the wild |
|
natural landscape |
The natural scenery and other attractions in the surrounding area are also important factors that attract tourists; the better these landscapes, the higher the value of the area’s ecological services. |
1 = worse natural landscape 2= status quo 3= better natural landscape |
|
infrastructure and facilities |
Food and beverage services, accommodation, and road conditions in the area also affect tourists’ decision to visit; the better these facilities, the higher the value of the scenic location. |
1 = worse development of infrastructure facilities 2 = status quo 3 = better development of infrastructure facilities |
|
ticket price |
Tickets prices have a negative effect; the marginal substitution rate with other attributes is negative. |
1 = 18.13 USD 2 = 20.40 USD (status quo) 3 = 22.67 USD 4 = 24.93 USD |
* The amount is converted according to the average exchange rate in 2018: 1 US dollar to 6.6174 yuan (All dollar conversions in this paper are based on this exchange rate).
Table 2. Examples of selection set in questionnaire.
|
|
Wild animals |
Natural landscape |
Infrastructure and facilities |
Ticket price |
I choose |
|
Route 1 (Status quo) |
see wild animals in captivity |
status quo |
status quo |
18.13 USD |
□ |
|
Route 2 (Improvement plan) |
see wild animals in captivity |
status quo |
better development of infrastructure facilities |
22.67 USD |
□ |
|
Route 3 (Improvement plan) |
cannot see wild animals |
better natural landscape |
status quo |
20.40 USD |
□ |
Comment 3: The figure 2 is very difficult to understand without detailed description in the figure caption (which should be in the same page).
Response 3: Thank you for your comment and we agree with you. We have revised the figure caption into “The main source of tourists in Huayang Town”.
As the paper has been revised many times, some traces of revision may not be preserved, but all the revisions have been reflected in the current final revision, please understand.
Thanks again for all the comments from reviewers and the editor. We have very carefully discussed and revised the suggestions of all reviewers. Hope for your reply.

Reviewer 2 Report
This is a well written manuscript that highlights the importance of wildlife eco-tourism in natural areas.
Overall, I think that even though it was not the main goal of the manuscript, the authors did not properly discuss the consequences of ecotourism on wildlife conservation, in light of the results of the survey. How the ecotourism might impact on local wildlife and wildlife conservation? How is the people perception on the cultural value of wild animals influencing their conservation?
The authors evaluated only possible ways of exploiting wildlife and natural resources as ecosystem services provided by nature to the society.
Below, you will find some suggestions to reshape Introduction and Discussion sections according to new references that have not been appointed by the authors.
According to Isaacs, “Ecotourism is a proxy market designed to align consumers' preferences for recreation with the protection of environmental assets. Because it does not necessarily address the direct protection of those assets, it is prone to market failure. Pressures on governments and firms involved in providing ecotourism services will impair their ability to minimize detrimental effects of human economic behavior. Ethical appeals to minimize harmful practices face serious obstacles. Promoting ecotourism may actually distract from more appropriate means of environmental protection.” (Isaacs, Jack Coburn. "The limited potential of ecotourism to contribute to wildlife conservation." Wildlife society bulletin 28.1 (2000): 61-69)
Moreover, “Despite the “win-win” idea, scholars and practitioners debate the meaning and merits of ecotourism.” Stronza, Amanda L., Carter A. Hunt, and Lee A. Fitzgerald. "Ecotourism for conservation?." Annual Review of Environment and Resources 44 (2019): 229-253.
And, “It is pertinent to reflect on the reasons behind the level of importance of a given species within a community. The indicators of significance and, more importantly the causes for this valuation, respond to multifactorial processes. A greater or lesser Cultural Significance can be assigned based on diverse factors, both those linked to the particular cultural features of the human group assigning it, and the intrinsic features of the species. Furthermore, these factors are shaped and re-shaped by historic processes.” García del Valle, Y., Naranjo, E.J., Caballero, J. et al. Cultural significance of wild mammals in mayan and mestizo communities of the Lacandon Rainforest, Chiapas, Mexico. J Ethnobiology Ethnomedicine 11, 36 (2015).
The main tool that has been used by the authors in this study is the questionnaire that is not present in the text. I suggest to add the questionnaire in the supplementary materials as the readers might benefit from reading it and it might also integrate the comprehension of the work.
I think the manuscript might be accepted after major revisions that need to be appointed.
Please find below the points raised line by line.
44-47: “Owing to a lack of trading markets and the difficulty of defining the property rights to these products or services (e.g. air quality, biodiversity, and climate-regulating ecosystem services), natural resources have always been over-exploited [5]”. This is a strong assumption and you should provide more than one reference, which is actually related to a restricted area.
53-54: This sentence needs to be referenced
53-66: this is a very long background with too many examples that should be summarized by the authors
71: “value evaluation” sounds difficult to read, it might be better to find another term
73-74: this sentence needs a reference
133-135 this sentence needs to be referenced
162: “It is also one of the areas with the richest 162 biodiversity in China”: to which animal/plant species are you referring to? You should at least add a reference.
281: attributes
284-286: a variance inflation factors (VIF) < 10 is quite high threshold. You should reconsider some of your results according to this study: https://besjournals.onlinelibrary.wiley.com/doi/pdf/10.1111/j.2041-210X.2009.00001.x “A protocol for data exploration to avoid common statistical problems”
306: Table 5 might be added to the supplementary materials
341-344: This is a strong assumption with cultural and religious implications that are not developed enough in the manuscript. “In the survey, it was found that religious tourists believe that protecting wild animals is a natural pursuit and that they do not need to pay extra money to do so.” To which religion belonged the tourists that believed it? Are you sure that their willingness to pay a ticket to visit the natural area is only influenced/correlated to their religion?
“They believe that protecting wild animals has nothing to do with money”. Why do you need to justify the beliefs of a religion (especially if you did not specify which kind of religion)?
“This is the result of the special ideological influence of certain religions practiced in Chinese.” You need to reference this statement. What do you mean by “special ideological influence”? And what are the “certain religions practiced in China”?
417: foreigners or non-local tourists?
Author Response
We appreciate the comments from the reviewer. We have itemized all comments and addressed them individually as follows.
Comment 1: Overall, I think that even though it was not the main goal of the manuscript, the authors did not properly discuss the consequences of ecotourism on wildlife conservation, in light of the results of the survey. How the ecotourism might impact on local wildlife and wildlife conservation? How is the people perception on the cultural value of wild animals influencing their conservation?
The authors evaluated only possible ways of exploiting wildlife and natural resources as ecosystem services provided by nature to the society.
Below, you will find some suggestions to reshape Introduction and Discussion sections according to new references that have not been appointed by the authors.
Response 1: Thank you very much for your comments. We refer to the references given by the reviewers, and modify the introduction and discussion as follows:
L42-44. A greater or lesser cultural significance can be assigned based on diverse factors, both those linked to the particular cultural features of the human group assigning it and the intrinsic features of the species [4].
L45-50. Ecotourism is a specific form of natural ecosystem providing services for human beings. It can improve the well-being of local community residents and tourists[5], however, there is still uncertainty about whether ecotourism can improve the effect of ecological protection[6]. Some studies have shown that tourists' perceived value of the ecological environment has a certain impact on their protection behavior[7]. Therefore, it is of certain significance to study the value evaluation of tourist destinations in the process of ecotourism.
L281-286. According to the survey results, 78.07% of the tourists, especially those with children, travel to the area for the purpose of seeing wild animals. Most of them said in the interview that through this tour, they hope their children can become more familiar with some of the wild animals and have an educational experience. Therefore, wildlife is the most important factor to attract tourists to Huayang Town. The results further illustrate that wildlife has played a certain role in promoting ecotourism, and the protection of wildlife has brought certain social benefits [6].
L324-328 .This result shows that although people participate in ecotourism mainly to see wild animals, there is also high demand for other entertainment facilities. While this can lead to the development of other ecotourism attractions, it does not necessarily address the direct protection of those assets, as it is prone to market failure [50].
Comment 2: The main tool that has been used by the authors in this study is the questionnaire that is not present in the text. I suggest to add the questionnaire in the supplementary materials as the readers might benefit from reading it and it might also integrate the comprehension of the work.
Response 2: Thank you very much for your comments.We added the questionnaire in the appendix.
Appendix A
Tourist Questionnaire (A)
- Please answer the questions in the form below
|
Question |
Answer |
|
What is your gender? |
1: Male;2: Female |
|
How old are you? |
1: ≤19;2: 20–29;3: 30–39;4: 40–49;5: 50–59;6: ≥60 |
|
What is your education level? |
1: Primary school and below; 2: Junior middle school; 3: High school (vocational); 4: Speciality; 5: Undergraduate; 6: Masters and above |
|
What is your monthly income (RMB)? |
1: ≤301; 2: 302-604; 3: 605-907; 4: 908-1209; 5: 1210-1511; 6: 1512-2267; 7: 2268-3022; 8:≥20000. |
|
Do you have any religious beliefs |
1: None;2: Buddhism; 3: Christianity; 4: Islam; 5: Taoism; 6:Other |
|
Where are you coming from? |
Province________, City________, County________. |
- What is the main purpose of your trip?
1=Scenic sightseeing; 2=Perception of cultural customs; 3=Wildlife viewing;
4=Visit to Historical Sites; 5=Scientific research; 6=Religious pilgrimage;
7=Relax with your family; 8=others
- Can you identify the local wildlife? 、 、 、 、 .
- Are you more willing to protect wildlife after travelling here?1=Yes; 0=No.
- Tables 1-6 assume that you will encounter three scenarios in the scenic spot. Which options would increase your level of satisfaction with this tour?
|
Table 1 |
|
|
|
|
|
Wild animals |
Natural landscape |
Infrastructure and facilities |
Ticket price |
I choose |
|
see wild animals in captivity |
status quo |
status quo |
18.13 USD |
□ |
|
see wild animals in captivity |
status quo |
better infrastructure facilities |
22.67 USD |
□ |
|
cannot see wild animals |
better natural landscape |
status quo |
20.40 USD |
□ |
|
|
|
|
|
|
|
Table2 |
|
|
|
|
|
Wild animals |
Natural landscape |
Infrastructure and facilities |
Ticket price |
I choose |
|
see wild animals in captivity |
worse natural landscape |
better infrastructure facilities |
20.40 USD |
□ |
|
cannot see wild animal |
status quo |
status quo |
18.13 USD |
□ |
|
see wild animals in captivity |
status quo |
status quo |
18.13 USD |
□ |
|
Table3 |
|
|
|
|
|
Wild animals |
Natural landscape |
Infrastructure and facilities |
Ticket price |
I choose |
|
see wild animals in captivity |
worse natural landscape |
better infrastructure facilities |
20.40 USD |
□ |
|
see animals in the wild |
better natural landscape |
status quo |
22.67 USD |
□ |
|
see wild animals in captivity |
status quo |
status quo |
18.13 USD |
□ |
|
|
|
|
|
|
|
Table4 |
|
|
|
|
|
Wild animals |
Natural landscape |
Infrastructure and facilities |
Ticket price |
I choose |
|
cannot see wild animals |
better natural landscape |
status quo |
20.40 USD |
□ |
|
see animals in the wild |
better natural landscape |
better infrastructure facilities |
22.67 USD |
□ |
|
see wild animals in captivity |
status quo |
status quo |
18.13 USD |
□ |
|
|
|
|
|
|
|
Table5 |
|
|
|
|
|
Wild animals |
Natural landscape |
Infrastructure and facilities |
Ticket price |
I choose |
|
see wild animals in captivity |
status quo |
status quo |
20.40 USD |
□ |
|
see animals in the wild |
better natural landscape |
better infrastructure facilities |
24.93 USD |
□ |
|
see wild animals in captivity |
status quo |
status quo |
18.13 USD |
□ |
|
|
|
|
|
|
|
Table6 |
|
|
|
|
|
Wild animals |
Natural landscape |
Infrastructure and facilities |
Ticket price |
I choose |
|
see wild animals in captivity |
status quo |
status quo |
20.40 USD |
□ |
|
see animals in the wild |
worse natural landscape |
better infrastructure facilities |
22.67 USD |
□ |
|
see wild animals in captivity |
status quo |
status quo |
18.13 USD |
□ |
*There are also six selection sets in questionnaire B.
Comment 3: 44-47: “Owing to a lack of trading markets and the difficulty of defining the property rights to these products or services (e.g. air quality, biodiversity, and climate-regulating ecosystem services), natural resources have always been over-exploited [5]”. This is a strong assumption and you should provide more than one reference, which is actually related to a restricted area.
Response 3: Thank you very much and we agree with you. We have add 3 references to support this assumption in the text.
Wade, R. The management of common property resources: Collective action as an alternative to privatisation or state regulation. Camb J Econ 1987, 11(2), 95-106.
Kaul, I.; Grungberg, I.; Stern, M.A. Global Public Goods Concepts, Policies and Strategies. In Global Public Goods: International Cooperation in the 21st Century, Kaul, I., Grungberg, I., Stern, M.A., Eds.; Oxford University Press: Oxford, United Kingdom, 1999, p. 450.
Kocher, M.G.; Tan, F.; Yu, J. Providing global public goods: Electoral delegation and cooperation. Econ Inq 2018, 56(1), 381-397
Comment 4: 53-54: This sentence needs to be referenced
Response 4:Thank you very much and we agree with you. We have add 3 references to support this assumption as follows:
Diafas, I.; Barkmann, J.; Mburu, J. Measurement of bequest value using a non-monetary payment in a choice experiment—The case of improving forest ecosystem services for the benefit of local communities in rural Kenya. Ecol Econ 2017, 140, 157-165.
Birol, E.; Karousakis, K.; Koundouri, P. Using a choice experiment to account for preference heterogeneity in wetland attributes: The case of Cheimaditida wetland in Greece. Ecol Econ 2006, 60(1), 145-156.
Oleson, K.; Barnes, M.; Brander, L.; Oliver, T.; van Beek, I.; Zafindrasilivonona, B.; van Beukering, P. Cultural bequest values for ecosystem service flows among Indigenous fishers: A discrete choice experiment validated with mixed methods. Ecol Econ 2015, 114, 104–116.
Comment 5 :53-66: this is a very long background with too many examples that should be summarized by the authors
Response 5 : We agree with your opinion and summarized as follows:
Hanley et al. studied the factors influencing the different needs of climbing sites in Scotland using four attributes [19].
Comment 6: 71: “value evaluation” sounds difficult to read, it might be better to find another term
Response 6: We agree with your comment and we use “Value Estimatio” instead.
Comment 7: 73-74: this sentence needs a reference
Response 7: Thank you for your comment and we have added a reference in the sentence as follow:
Kainuma, Y.; Tawara, N. A multiple attribute utility theory approach to lean and green supply chain management. Int J of Prod Econ 2006, 101(1), 99-108.
Comment 8: 133-135 this sentence needs to be referenced
Response 8: Thank you for your comment and we have added a reference in the sentence as follow:
40.Lancsar, E.; Louviere, J. Conducting discrete choice experiments to inform healthcare decision making. Pharmacoeconomics 2008, 26(8), 661-677.
Rose, J.M.; Bliemer, M.C.J.; Hensher, D.A., et al. Designing efficient stated choice experiments in the presence of reference alternatives. Transport Res B-Meth 2008, 42(4), 395-406.
Comment 9: 162: “It is also one of the areas with the richest 162 biodiversity in China”: to which animal/plant species are you referring to? You should at least add a reference.
Response 9: Thank you for your comments and we revised it as follows:
It is also one of the areas with the richest species diversity and ecosystem in China [5].
Comment 10: 284-286: a variance inflation factors (VIF) < 10 is quite high threshold. You should reconsider some of your results according to this study: https://besjournals.onlinelibrary.wiley.com/doi/pdf/10.1111/j.2041-210X.2009.00001.x “A protocol for data exploration to avoid common statistical problems”
Response 10: Thank you for your comment and we read the paper carefully. Generally, if ariance inflation factors(VIF) is greater than 10, there will be serious collinearity(Montgomery & Peck ,1992). According to paper given by the reviewer, “ High, or even moderate, collinearity is especially problematic when ecological signals are weak. In that case, even low VIF may cause nonsignificant parameter estimates, compared to the situation without collinearity”. In our study, this situation is not existed, so VIF< 10 is acceptable.
Montgomery, D.C.; Peck, E.A. Introduction to Linear Regression Analysis; Wiley: New York, USA, 1992.
Comment 11: 306: Table 5 might be added to the supplementary materials
Response 11: Thank you for yourcomment. Table 5 is the empirical estimation results of Model 1 and Model 2. Supplementary materials are under the table as follows:
- The regression result of the attribute variable is not added to the interactions, so the marginal coefficients can be explained directly. (2) Considering the number of parameters in one regression and to reduce collinearity, the interactions are included in the explanatory variables. (3) The sign of the estimated standard deviations is irrelevant; they should be interpreted as being positive. (4) ***, **, and * denote statistical significance at the 1%, 5%, and 10% levels, respectively.
Comment 12 : 341-344: This is a strong assumption with cultural and religious implications that are not developed enough in the manuscript. “In the survey, it was found that religious tourists believe that protecting wild animals is a natural pursuit and that they do not need to pay extra money to do so.” To which religion belonged the tourists that believed it? Are you sure that their willingness to pay a ticket to visit the natural area is only influenced/correlated to their religion?
“They believe that protecting wild animals has nothing to do with money”. Why do you need to justify the beliefs of a religion (especially if you did not specify which kind of religion)?
“This is the result of the special ideological influence of certain religions practiced in Chinese.” You need to reference this statement. What do you mean by “special ideological influence”? And what are the “certain religions practiced in China”?
Response 12: Thank you very much for your comments. We agree with your comment and revised in the text as follows:
L363-369. In the survey, it was found that Buddhist tourists believe that there are many immortals found in nature, even wild animals are believed to be an incarnation of immortals, so they think that human beings should take the initiative to protect wild animals. In their view, these are things that must be done and should not be protected by the way of payment, which is a kind of remedial action after the event. This is the result of the special ideological influence of Buddhism practised in Chinese. Therefore, the Buddhist tourists’ willingness-to-pay is a little bit lower than others.
Comment 13 :417: foreigners or non-local tourists?
Response 13: We agree with your comment and we use “Value Estimatio” instead of “foreigner”.
As the paper has been revised many times, some traces of revision may not be preserved, but all the revisions have been reflected in the current final revision, please understand.
Thanks again for all the comments from reviewers and the editor. We have very carefully discussed and revised the suggestions of all reviewers. Hope for your reply.

Round 2
Reviewer 2 Report
Thank you for addressing all the comments raised.
I think that the manuscript can be now accepted for publication.